# Computing high-dimensional optimal transport by flow neural networks

## Abstract

Flow-based models are widely used in generative tasks, including normalizing flow, where a neural network transports from a data distribution $P$ to a normal distribution. This work develops a flow-based model that transports from $P$ to an arbitrary $Q$ where both distributions are only accessible via finite samples. We propose to learn the dynamic optimal transport between $P$ and $Q$ by training a flow neural network. The model is trained to find an invertible transport map between $P$ and $Q$ optimally by minimizing the transport cost. The trained optimal transport flow allows for performing many downstream tasks, including infinitesimal density ratio estimation and distribution interpolation in the latent space for generative models. The effectiveness of the proposed model on high-dimensional data is empirically demonstrated in mutual information estimation, energy-based generative models, and image-to-image translation.

## 1 Introduction

The problem of finding a transport map between two general distributions $P$ and $Q$ in high dimension is essential in statistics, optimization, and machine learning. When both distributions are only accessible via finite samples, the transport map needs to be learned from data. In spite of the modeling and computational challenges, this setting has applications in many fields. For example, transfer learning in domain adaption aims to obtain a model on the target domain at a lower cost by making use of an existing pre-trained model on the source domain (Courty et al., 2014; 2017), and this can be achieved by transporting the source domain samples to the target domain using the transport map. The (optimal) transport has also been applied to achieve model fairness (Silvia et al., 2020). By transporting distributions corresponding to different sensitive attributes to a common distribution, an unfair model is calibrated to match certain desired fairness criteria (e.g., demographic parity (Jiang et al., 2020)). The transport map can also be used to provide intermediate interpolating distributions between $P$ and $Q$. In density ratio estimation (DRE), this bridging facilitates the so-called "telescopic" DRE (Rhodes et al., 2020) which has been shown to be more accurate when $P$ and $Q$ significantly differ. Furthermore, learning such a transport map between two sets of images can facilitate solving problems in computer vision, such as image restoration and image-to-image translation (Isola et al., 2017).

This work focuses on a continuous-time formulation of the problem where we are to find an invertible transport map $T_t : \mathbb{R}^d \to \mathbb{R}^d$ continuously parametrized by time $t \in [0, 1]$ and satisfying that $T_0 = \text{Id}$ (the identity map) and $(T_1)_{\#} P = Q$. Here we denote by $T_{\#} P$ the push-forward of distribution $P$ by a mapping $T$, such that $(T_{\#} P)(\cdot) = P(T^{-1}(\cdot))$. Suppose $P$ and $Q$ have densities $p$ and $q$ respectively in $\mathbb{R}^d$ (we also use the push-forward notation $\#$ on densities), the transport map $T_t$ defines

$$\rho(x, t) := (T_t)_{\#} p, \quad \text{s.t.} \quad \rho(x, 0) = p, \quad \rho(x, 1) = q.$$

We will adopt the neural Ordinary Differential Equation (ODE) approach Chen et al. (2018) where we represent $T_t$ as the solution map of an ODE, which is further parametrized by a continuous-time residual network. The resulting map $T_t$ is invertible, and the inversion can be computed by integrating the neural ODE reverse in time. Our model learns the flow from two sets of finite samples from $P$ and $Q$. The velocity field in the neural ODE will be optimized to minimize the transport cost so as to approximate the optimal velocity in dynamic optimal transport (OT) formulation, i.e. Benamou-Brenier equation.

The neural-ODE model has been intensively developed in Continuous Normalizing Flows (CNF) Kobyzev et al. (2020). In CNF, the continuous-time flow model, usually parametrized by a neural ODE, transports from a data distribution $P$ (accessible via finite samples) to a terminal analytical distribution which is typically the normal one $\mathcal{N}(0, I_d)$, per the name "normalizing". The study of normalizing flow dated back to non-deep models with statistical applications (Tabak & Vanden-Eijnden, 2010), and deep CNFs have recently developed into a popular tool for generative models and likelihood inference of high dimensional data. CNF models rely on the analytical expression of the terminal distribution in training. Since our model is also a flow model that transports from data distribution $P$ to a general (unknown) data distribution $Q$, both accessible via empirical samples, we name our model "Q-flow" which is inspired by the CNF literature.

In summary, the contributions of the work include:

- We develop a flow-based model *Q-flow* net to learn a continuous invertible optimal transport map between arbitrary pair of distributions $P$ and $Q$ in $\mathbb{R}^d$ from two sets of samples of the distributions. We propose to train a neural ODE model to minimize the transport cost such that the flow approximates the optimal transport in dynamic OT. The end-to-end training of the model refines an initial flow that may not attain the optimal transport, e.g., obtained by training two CNFs or other interpolating schemes.

- Leveraging the trained optimal transport Q-flow net, we propose a new DRE approach by training a separate continuous-time neural network using classification losses along the time grid. The proposed DRE method improves the performance in high dimension, demonstrated by high-dimensional mutual information estimation and energy-based generative models.

- We show the effectiveness of the approach on simulated and real data. On the image-to-image translation task, our Q-flow gradually transforms an input image to a target one that resembles in style and achieves competitive quantitative metrics against the baselines.

## 1.1 RELATED WORKS

**Normalizing flows.** When the target distribution $Q$ is an isotropic Gaussian $\mathcal{N}(0, I_d)$, normalizing flow models have demonstrated vast empirical successes in building an invertible transport $T_t$ between $P$ and $\mathcal{N}(0, I_d)$ (Kobyzev et al., 2020). The transport is parametrized by deep neural networks, whose parameters are trained via minimizing the KL-divergence between transported distribution $(T_1)_\# P$ and $\mathcal{N}(0, I_d)$. Various continuous (Grathwohl et al., 2019; Finlay et al., 2020) and discrete (Dinh et al., 2016; Behrmann et al., 2019) normalizing flow models have been developed, along with proposed regularization techniques (Onken et al., 2021; Xu et al., 2022a;b) that facilitate the training of such models in practice.

Since our Q-flow is in essence a transport-regularized flow between $P$ and $Q$, we further review related works on building normalizing flow models with transport regularization. (Finlay et al., 2020) trained the flow trajectory with regularization based on $\ell_2$ transport cost and Jacobian norm of the network-parametrized velocity field. (Onken et al., 2021) proposed to regularize the flow trajectory by $\ell_2$ transport cost and the deviation from the HJB equation. These regularization have shown to effectively improve over un-regularized models at a reduced computational cost. Regularized normalizing flow models have also been used to solve high dimensional Fokker-Planck equations (Liu et al., 2022) and mean-field games (Huang et al., 2023).

**Distribution interpolation by neural networks.** Recently, there have been several works establishing a continuous-time interpolation between general high-dimensional distributions. (Albergo & Vanden-Eijnden, 2023) proposed to use a stochastic interpolant map between two arbitrary distributions and train a neural network parametrized velocity field to transport the distribution along the interpolated trajectory. (Neklyudov et al., 2023) proposed an action matching scheme that leverages a pre-specified trajectory between $P$ and $Q$ to learn the OT map between two *infinitesimally close* distributions along the trajectory. (Liu, 2022) proposed rectified flow which starts from an initial coupling of $P$ and $Q$ and iteratively rectifies it to converge to the optimal coupling. Same as in (Albergo & Vanden-Eijnden, 2023; Neklyudov et al., 2023; Lipman et al., 2023), our neural-ODE based approach also computes a deterministic probability transport map, in contrast to SDE-based diffusion models (Song et al., 2021). Notably, the interpolant mapping used in these prior works

is generally not the optimal transport interpolation. In comparison, our proposed Q-flow optimizes the interpolant mapping parametrized by a neural ODE and approximates the optimal velocity in dynamic OT (see Section 2). Generally, the flow attaining optimal transport can lead to improved model efficiency and generalization performance Huang et al. (2023). In this work, the proposed method aims to solve the dynamic OT trajectory by a flow network, and we experimentally show that the optimal transport flow benefits high-dimensional DRE and image-to-image translation.

**Optimal transport between general distributions.** The problem of OT dates back to the work by Gaspard Monge (Monge, 1781), and since then many mathematical theories and computational tools have been developed to tackle the question (Villani et al., 2009; Benamou & Brenier, 2000; Peyré et al., 2019). Several works have attempted to make computational OT scalable to high dimensions, including (Lavenant et al., 2018) which applied convex optimization using Riemannian structure of the space of discrete probability distributions, and (Lee et al., 2021) by $L^1$ and $L^2$ versions of the generalized unnormalized OT solved by Nesterov acceleration. Several deep approaches have also been developed recently. (Coeurdoux et al., 2023) leveraged normalizing flow to learn an approximate transport map between two distributions from finite samples, where the flow model has a restricted architecture and the OT constraint is replaced with sliced-Wasserstein distance which may not computationally scale to high dimensional data. Several works have also considered casting the optimal transport problem into a minimax problem based on either the Kantorovich formulation of (Xie et al., 2019; Korotin et al., 2023) or the Monge formulation (Fan et al., 2022). In comparison, our approach computes the continuous-time dynamic OT mapping parametrized by the optimal velocity field, which directly provides a continuous interpolation between two distributions and can be applied to tasks like DRE.

## 2 PRELIMINARIES

**Neural ODE and CNF.** Neural ODE Chen et al. (2018) parametrized an ODE in $\mathbb{R}^d$ by a residual network. Specifically, let $x(t)$ be the solution of

$$\dot{x}(t) = f(x(t), t; \theta), \quad x(0) \sim p. \tag{1}$$

where $f(x, t; \theta)$ is a velocity field parametrized by the neural network. Since we impose a distribution $P$ on the initial value $x(0)$, the value of $x(t)$ at any $t$ also observes a distribution $p(x, t)$ (though $x(t)$ is deterministic given $x(0)$). In other words, $p(\cdot, t) = (T_t)_{\#}p$, where $T_t$ is the solution map of the ODE, namely $T_t(x) = x + \int_0^t f(x(s), s; \theta)ds$, $x(0) = x$. In the context of CNF (Kobyzev et al., 2020), the training of the flow network $f(x, t; \theta)$ is to minimize the KL divergence between the terminal density $p(x, T)$ at some $T$ and a target density $p_Z$ which is the normal distribution. The computation of the objective relies on the expression of normal density and can be estimated on finite samples of $x(0)$ drawn from $p$.

**Dynamic OT (Benamou-Brenier).** The Benamou-Brenier equation below provides the dynamic formulation of OT Villani et al. (2009); Benamou & Brenier (2000)

$$\inf_{\rho, v} \mathcal{T} := \int_0^1 \mathbb{E}_{x(t) \sim \rho(\cdot, t)} \|v(x(t), t)\|^2 dt \tag{2}$$
$$s.t. \quad \partial_t \rho + \nabla \cdot (\rho v) = 0, \quad \rho(x, 0) = p(x), \quad \rho(x, 1) = q(x),$$

where $v(x, t)$ is a velocity field and $\rho(x, t)$ is the probability mass at time $t$ satisfying the continuity equation with $v$. The action $\mathcal{T}$ is the transport cost. Under regularity conditions of $p, q$, the minimum $\mathcal{T}$ in (2) equals the squared Wasserstein-2 distance between $p$ and $q$, and the minimizer $v(x, t)$ can be interpreted as the optimal control of the transport problem.

## 3 LEARNING DYNAMIC OT BY Q-FLOW NETWORK

We introduce the formulation and training objective of the proposed OT Q-flow net in Section 3.1. The training technique consists of the end-to-end training (Section 3.2) and the construction of the initial flow (Section 3.3).

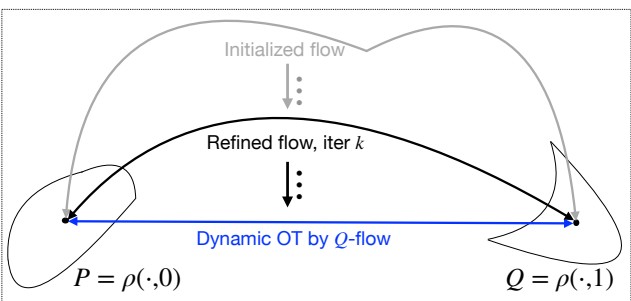

Figure 1: Illustration of learning the dynamic OT using our Q-flow (blue), which invertibly transports between $P$ and $Q$ over the interval $[0, 1]$ with the least transport cost. Taking any initial flow (grey) between $P$ and $Q$, we iteratively refine flow trajectories to obtain flows with smaller transport cost (black), converging gradually to the dynamic OT between these two distributions.

### 3.1 FORMULATION AND TRAINING OBJECTIVE

Given two sets of samples $\boldsymbol{X} = \{X_i\}_{i=1}^N$ and $\tilde{\boldsymbol{X}} = \{\tilde{X}_j\}_{j=1}^M$, where $X_i \sim P$ and $\tilde{X}_j \sim Q$ i.i.d., we train a neural ODE model $f(x, t; \theta)$ (1) to represent the transport map $T_t$. The formulation is symmetric from $P$ to $Q$ and vice versa, and the loss will also have symmetrically two parts. We call $P \to Q$ the forward direction and $Q \to P$ the reverse direction.

Our training objective is based on the dynamic OT (2) on time $[0, 1]$, where we solve the velocity field $v(x, t)$ by $f(x, t; \theta)$. The terminal condition $\rho(\cdot, 1) = q$ is relaxed by a KL divergence (see, e.g., (Ruthotto et al., 2020)). The training loss in forward direction is written as

$$\mathcal{L}^{P \to Q} = \mathcal{L}_{\mathrm{KL}}^{P \to Q} + \gamma \mathcal{L}_T^{P \to Q}, \tag{3}$$

where $\mathcal{L}_{\mathrm{KL}}$ represents the relaxed terminal condition and $\mathcal{L}_T$ is the Wasserstein-2 transport cost to be specified below; $\gamma > 0$ is a weight parameter, and with small $\gamma$ the terminal condition is enforced.

**KL loss.** Now we specify the first term in the loss (3) $\mathcal{L}_{\mathrm{KL}}^{P \to Q}$. We define the solution mapping of (1) from $s$ to $t$ as

$$T_s^t(x; \theta) = x(s) + \int_s^t f(x(t'), t'; \theta) dt', \tag{4}$$

which is also parametrized by $\theta$, and we may omit the dependence below. By the continuity equation in (2), $\rho(\cdot, t) = (T_0^t)_{\#} p$. The terminal condition $\rho(\cdot, 1) = q$ is relaxed by minimizing

$$\mathrm{KL}(p_1 || q) = \mathbb{E}_{x \sim p_1} \log(p_1(x)/q(x)), \quad p_1 := (T_0^1)_{\#} p.$$

The expectation $\mathbb{E}_{x \sim p_1}$ is estimated by the sample average over $(X_1)_i$ which observes density $p_1$ i.i.d., where $(X_1)_i := T_0^1(X_i)$ is computed by integrating the neural ODE from time 0 to 1.

It remains to have an estimator of $\log(p_1/q)$ to compute $\mathrm{KL}(p_1 || q)$, and we propose to train a logistic classification network $r_1(x; \varphi_r)$ with parameters $\varphi_r$ for this. The inner-loop training of $r_1$ is by

$$\min_{\varphi_r} \frac{1}{N} \sum_{i=1}^N \log(1 + e^{r_1(T_0^1(X_i; \theta); \varphi_r)}) + \frac{1}{M} \sum_{j=1}^M \log(1 + e^{-r_1(\tilde{X}_j; \varphi_r)}). \tag{5}$$

The functional optimal $r_1^*$ of the population version of loss (5) equals $\log(q/p_1)$ by direct computation, and as a result, $\mathrm{KL}(p_1 || q) = -\mathbb{E}_{x \sim p_1} r_1^*(x)$. Now take the trained classification network $r_1$ with parameter $\hat{\varphi}_r$, we can estimate the finite sample KL loss as

$$\mathcal{L}_{\mathrm{KL}}^{P \to Q}(\theta) = -\frac{1}{N} \sum_{i=1}^N r_1(T_0^1(X_i; \theta); \hat{\varphi}_r), \tag{6}$$

where $\hat{\varphi}_r$ is the computed minimizer of (5) solved by inner loops. In practice, when the density $p_1$ is close to $q$, the DRE by training classification net $r_1$ can be efficient and accurate. We will apply the minimization (5) after the flow net is properly initialized which guarantees the closeness of $p_1 = (T_0^1)_{\#} p$ and $q$ to begin with.

$W_2$ **regularization.** Now we specify the second term in the loss (3) that defines the Wasserstein-2 regularization. To compute the transport cost $\mathcal{T}$ in (2) with velocity field $f(x, t; \theta)$, we use a time grid on $[0, 1]$ as $0 = t_0 < t_1 < \ldots < t_K = 1$. The choice of the time grid is algorithmic (since the flow model is parametrized by $\theta$ throughout time) and may vary over experiments, see more details in Section 3.2. Define $h_k = t_k - t_{k-1}$, and $X_i(t; \theta) := T_0^t(X_i; \theta)$, the $W_2$ regularization is written as

$$\mathcal{L}_T^{P \to Q}(\theta) = \sum_{k=1}^K \frac{1}{h_k} \left( \frac{1}{N} \sum_{i=1}^N \|X_i(t_k; \theta) - X_i(t_{k-1}; \theta)\|^2 \right). \tag{7}$$

It can be viewed as a time discretization of $\mathcal{T}$. Meanwhile, since (omitting dependence on $\theta$) $X_i(t_k) - X_i(t_{k-1}) = T_{t_{k-1}}^{t_k}(X_i(t_{k-1}))$, the population form of (7) $\sum_{k=1}^K \mathbb{E}_{x \sim \rho(\cdot, t_{k-1})} \|T_{t_{k-1}}^{t_k}(x; \theta)\|^2 / h_k$ in minimization can be interpreted as the discrete-time summed (square) Wasserstein-2 distance (Xu et al., 2022a)

$$\sum_{k=1}^K W_2(\rho(\cdot, t_{k-1}), \rho(\cdot, t_k))^2 / h_k.$$

The $W_2$ regularization encourages a smooth flow from $P$ to $Q$ with small transport cost, which also guarantees the invertibility of the model in practice when the trained neural network flow approximates the optimal flow in (2).

**Flow in both directions.** To improve the numerical accuracy, we will design a training scheme that will take into account flow in both directions, $T_0^1$ and $T_1^0$; note that these transport maps are related to each other through (4). The formulation in the reverse direction is similar, where we transport $Q$-samples $\tilde{X}_i$ from 1 to 0 using the same neural ODE integrated in reverse time. Specifically, $\mathcal{L}^{Q \to P} = \mathcal{L}_{KL}^{Q \to P} + \gamma \mathcal{L}_T^{Q \to P}$, and $\mathcal{L}_{KL}^{Q \to P}(\theta) = -\frac{1}{M} \sum_{j=1}^M \tilde{r}_0(T_1^0(\tilde{X}_j; \theta); \hat{\varphi}_{\tilde{r}})$, where $\hat{\varphi}_{\tilde{r}}$ is obtained by inner-loop training of another classification net $\tilde{r}_0(x, \varphi_{\tilde{r}})$ with parameters $\varphi_{\tilde{r}}$ via

$$\min_{\varphi_{\tilde{r}}} \frac{1}{M} \sum_{j=1}^M \log(1 + e^{\tilde{r}_0(T_1^0(\tilde{X}_j; \theta); \varphi_{\tilde{r}})}) + \frac{1}{N} \sum_{i=1}^N \log(1 + e^{-\tilde{r}_0(X_i; \varphi_{\tilde{r}})}); \tag{8}$$

Define $\tilde{X}_j(t; \theta) := T_1^t(\tilde{X}_j; \theta)$, the reverse-time $W_2$ regularization is

$$\mathcal{L}_T^{Q \to P}(\theta) = \sum_{k=1}^K \frac{1}{h_k} \left( \frac{1}{M} \sum_{j=1}^M \|\tilde{X}_j(t_{k-1}; \theta) - \tilde{X}_j(t_k; \theta)\|^2 \right).$$

### 3.2 END-TO-END TRAINING ALGORITHM

In the end-to-end training, we assume that the Q-flow net has already been initiated as an approximate solution of the desired Q-flow, see more in Section 3.3. We then minimize $\mathcal{L}^{P \to Q}$ and $\mathcal{L}^{Q \to P}$ in an alternative fashion per "Iter", and the procedure is given in Algorithm 1. Hyperparameter choices and network architectures are further detailed in Appendix B.

**Time integration of flow.** In the losses (6) and (7), one need to compute the transported samples $X_i(t; \theta)$ and $\tilde{X}_j(t; \theta)$ on time grid points $\{t_k\}_{k=0}^K$. This calls for integrating the neural ODE on $[0, 1]$, which we conduct on a fine time grid $t_{k,s}$, $s = 0, \ldots, S$, that divides each subinterval $[t_{k-1}, t_k]$ into $S$ mini-intervals. We compute the time integration of $f(x, t; \theta)$ using a fixed-grid four-stage Runge-Kutta method on each mini-interval. The fine grid is used to ensure the numerical accuracy of ODE integration and the numerical invertibility of the Q-flow net, i.e., the error of using reverse-time integration as the inverse map (see inversion errors in Table A.1). It is also possible to first train the flow $f(x, t; \theta)$ on a time grid to warm start the later training on a refined grid, so as to improve convergence. We also find that the $W_2$ regularization can be computed at a coarser grid $t_k$ ($S$ is usually 3-5 in our experiments) without losing the effectiveness of Wasserstein-2 regularization. Finally, one can adopt an adaptive time grid, e.g., by enforcing equal $W_2$ movement on each subinterval $[t_{k-1}, t_k]$ Xu et al. (2022b), so that the representative points are more evenly distributed along the flow trajectory and the learning of the flow model can be further improved.

**Inner-loop training of $r_1$ and $\tilde{r}_0$.** Suppose the flow net has been successfully warm-started, the transported distributions $(T_0^1)_\# P \approx Q$ and $(T_1^0)_\# Q \approx P$. The two classification nets are first trained for $E_0$ epochs before the loops of training the flow model and then updated for $E_{\text{in}}$ inner-loop epochs in each outer-loop iteration. We empirically find that the diligent updates of $r_1$ and $\tilde{r}_0$ in lines 5 and 10 of Algorithm 1 are crucial for successful end-to-end training of Q-flow net. As we update the flow model $f(x, t; \theta)$, the push-forwarded distributions $(T_0^1)_\# P$ and $(T_1^0)_\# Q$ are conse-

---

**Algorithm 1** OT Q-flow refinement

**input** Pre-trained initial flow network $f(x(t), t; \theta)$; training data $\boldsymbol{X} \sim P$ and $\widetilde{\boldsymbol{X}} \sim Q$; hyperparameters: $\{\gamma, \{t_k\}_{k=1}^K, \texttt{Tot}, E, E_0, E_{\text{in}}\}$.
**output** Refined flow network $f(x(t), t; \theta)$
1: **for** Iter = $1, \ldots, \texttt{Tot}$ **do**
2:    (If Iter = 1) Train $r_1$ by minimizing (5) for $E_0$ epochs.
3:    **for** epoch = $, 1, \ldots, E$ **do** $\{\triangleright P \to Q \text{ refinement}\}$
4:       Update $\theta$ of $f(x(t), t; \theta)$ by minimizing $\mathcal{L}^{P \to Q}$.
5:       Update $r_1$ by minimizing (5) for $E_{\text{in}}$ epochs.
6:    **end for**
7:    (If Iter = 1) Train $\widetilde{r}_0$ by minimizing (8) for $E_0$ epochs.
8:    **for** epoch = $, 1, \ldots, E$ **do** $\{\triangleright Q \to P \text{ refinement}\}$
9:       Update $\theta$ of $f(x(t), t; \theta)$ by minimizing $\mathcal{L}^{Q \to P}$.
10:      Update $\widetilde{r}_0$ by minimizing (8) for $E_{\text{in}}$ epochs.
11:   **end for**
12: **end for**

---

quently changed, and then one will need to retrain $r_1$ and $\widetilde{r}_0$ timely to ensure an accurate estimate of the log-density ratio and consequently the KL loss. Compared with training the flow parameter $\theta$, the computational cost of the two classification nets is light which allows potentially a large number of inner-loop iterations if needed.

**Computational complexity.** We measure the computational complexity by the number of function evaluations of $f(x(t), t; \theta)$ and of the classification nets $\{r_1, \widetilde{r}_0\}$. Suppose the total number of epochs in outer loop training is $O(E)$, the dominating computational cost lies in the neural ODE integration, which takes $O(8KS \cdot E(M + N))$ function evaluations of $f(x, t; \theta)$. We remark that the Wasserstein-2 regularization (7) incurs no extra computation, since the samples $X_i(t_k; \theta)$ and $\tilde{X}_j(t_k; \theta)$ are available when computing the forward and reverse time integration of $f(x, t; \theta)$. The training of the two classification nets $r_1$ and $\widetilde{r}_0$ takes $O(4(E_0 + EE_{\text{in}})(M + N))$ additional evaluations of the two network functions since the samples $X_i(1; \theta)$ and $\tilde{X}_j(0; \theta)$ are already computed.

## 3.3 FLOW INITIALIZATION

We propose to initialize the Q-flow net by a flow model that approximately matches the transported distributions with the target distributions in both directions (and may not necessarily minimize the transport cost). Such an initialization will significantly accelerate the convergence of the end-to-end training, which can be viewed as a refinement of the initial flow.

The initial flow $f(x, t; \theta)$ may be specified using prior knowledge of the problem if available. Generally, when only two data sets $\boldsymbol{X}, \widetilde{\boldsymbol{X}}$ are given, the initial flow can be obtained by adopting existing methods in generative flows. In this work, we adopt two approaches: The first method is to construct the initial flow as a concatenation of two CNF models, each of which flows invertibly between $P$ and $Z$ and $Z$ and $Q$ for $Z \sim \mathcal{N}(0, I_d)$. Any existing neural-ODE CNF models may be adopted for this initialization (Grathwohl et al., 2019; Xu et al., 2022b). The second method adapts distribution interpolant neural networks. Specifically, one can use the linear interpolant mapping in (Rhodes et al., 2020; Choi et al., 2022; Albergo & Vanden-Eijnden, 2023) (see Appendix C), and train the neural network velocity field $f(x, t; \theta)$ to match the interpolation (Albergo & Vanden-Eijnden, 2023). Note that any other initialization scheme is compatible with the proposed end-to-end training of the Q-flow model to obtain the OT flow.

## 4 EXPERIMENTS

In this section, we demonstrate the effectiveness of the proposed method on several downstream tasks. The benefit of improving DRE between $P$ and $Q$ are shown in Sections 4.2–4.4, and the application to image-to-image translation is presented in Section 4.5. Additional ablation studies regarding hyper-parameter sensitivity are performed in Appendix B.5.

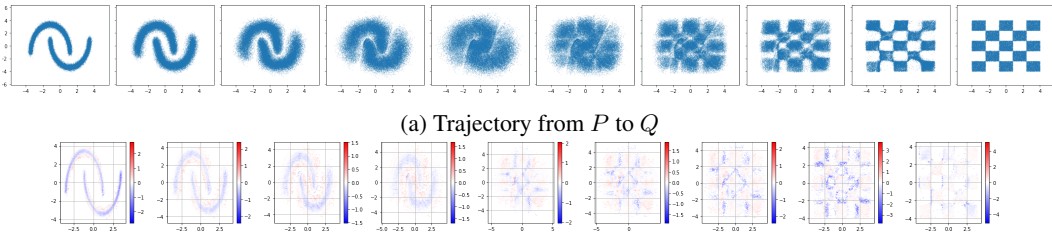

(a) Trajectory from $P$ to $Q$

(b) Estimated log-ratio between $P_{t_{k-1}}$ and $P_{t_k}$ by the trained Q-flow-ratio net.

Figure 2: Q-flow trajectory between arbitrary 2D distributions and corresponding log-ratio estimation. **Top**: intermediate distributions by Q-flow net. **Bottom**: corresponding log-ratio estimated by Q-flow-ratio net. Bluer color indicates smaller estimates of the difference $\log(p(x, t_k)/p(x, t_{k-1}))$ evaluated at the common support of the neighboring densities.

### 4.1 INFINITESIMAL DENSITY RATIO ESTIMATION (DRE)

For the DRE task, using the learned OT flow network between $P$ and $Q$, we propose to train a separate continuous-time neural network, called the *Q-flow-ratio* net, by minimizing a classification loss at time stamps along the flow trajectory. This differs from Choi et al. (2021) which used a 'time score matching' objective, and we also adopt a different time discretization. Details of the method are provided in Appendix A, see Algorithm A.2. In practice, we found our approach to train the density ratio network can be more efficient in some cases.

In the experimental results below, we denote our method as "Ours", and compare against three baselines of DRE in high dimensions. The baseline methods are: 1 ratio (by training a single classification network using samples from $P$ and $Q$), TRE (Rhodes et al., 2020), and DRE-$\infty$ (Choi et al., 2022). We denote $P_{t_k}$ with density $p(\cdot, t_k)$ as the pushforward distribution of $P$ by the Q-flow transport over the interval $[0, t_k]$. The set of distributions $\{P_{t_k}\}$ for $k = 1, \ldots, L$ builds a bridge between $P$ and $Q$.

### 4.2 TOY DATA IN 2D

**Gaussian mixtures.** We simulate $P$ and $Q$ as two Gaussian mixture models with three and two components, respectively, see additional details in Appendix B.1. We compute ratio estimates $\hat{r}(x)$ with the true value $r(x)$, which can be computed using the analytic expressions of the densities. The results are shown in Figure A.1. We see from the top panel that the mean absolute error (MAE) of Ours is evidently smaller than those of the baseline methods, and Ours also incurs a smaller maximum error $|\hat{r} - r|$ on test samples. This is consistent with the closest resemblance of Ours to the ground truth (first column) in the bottom panel. In comparison, DRE-$\infty$ tends to over-estimate $r(x)$ on the support of $Q$, while TRE and 1 ratio can severely under-estimate $r(x)$ on the support of $P$. As both the DRE-$\infty$ and TRE models use the linear interpolant scheme (16), the result suggests the benefit of training an optimal-transport flow for DRE.

**Two-moon to and from checkerboard.** We design two densities in $\mathbb{R}^2$ where $P$ represents the shape of two moons and $Q$ represents a checkerboard, see additional details in Appendix B.1. For this more challenging case, the linear interpolation scheme (16) creates a bridge between $P$ and $Q$ as shown in Figure A.4. The flow visually differs from the one obtained by the trained Q-flow net, as shown in Figure 2(a), and the latter is trained to minimize the transport cost. The result of Q-flow-ratio net is shown in Figure 2(b). The corresponding density ratio estimates of $\log p(x, t_k) - \log p(x, t_{k-1})$ visually reflect the actual differences in the two neighboring densities.

### 4.3 MUTUAL INFORMATION ESTIMATION FOR HIGH-DIMENSIONAL DATA

We evaluate different methods on estimating the mutual information (MI) between two correlated random variables from given samples. In this example, we let $P$ and $Q$ be two high-dimensional Gaussian distributions following the setup in (Rhodes et al., 2020; Choi et al., 2022), where we vary

the data dimension $d$ in the range of $\{40, 80, 160, 320\}$. Additional details can be found in Appendix B.2.

Figure 3 shows the results by different methods, where the baselines are trained under their proposed default settings. We find that the estimated MI by our method almost perfectly aligns with the ground truth MI values, reaching nearly identical performance as DRE-$\infty$ does. Meanwhile, Ours outperforms the other two baselines and the performance gaps increase as the dimension $d$ increases.

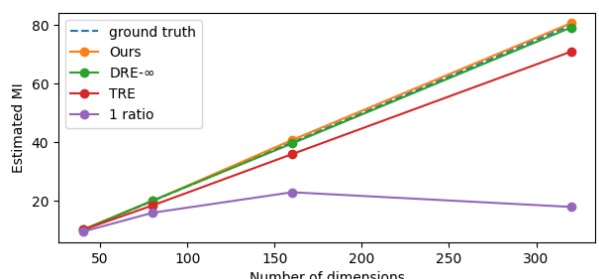

Figure 3: Estimated MI between two correlated high-dimensional Gaussian random variables.

## 4.4 Energy-based modeling of MNIST

We apply our approach in evaluating and improving an energy-base model (EBM) on the MNIST dataset (LeCun & Cortes, 2005). We follow the prior setup in (Rhodes et al., 2020; Choi et al., 2022), where $P$ is the empirical distribution of MNIST images, and $Q$ is the generated image distributions by three given pre-trained energy-based generative models: a Gaussian noise model, a Gaussian copula model, and a Rational Quadratic Neural Spline Flow model (RQ-NSF) (Durkan et al., 2019). Specifically, the images are in dimension $d = 28^2 = 784$, and each of the pre-trained models provides an invertible mapping $F : \mathbb{R}^d \to \mathbb{R}^d$, where $Q = F_\# \mathcal{N}(0, I_d)$. We train a Q-flow net between $(F^{-1})_\# P$ and $(F^{-1})_\# Q$, the latter by construction equals $\mathcal{N}(0, I_d)$. Using the trained Q-flow net, we go back to the input space and train the Q-flow-ratio net using the intermediate distributions between $P$ and $Q$. Additional details are in Appendix B.3.

The trained Q-flow-ratio $r(x, s; \hat{\theta}_r)$ provides an estimate of the data density $p(x)$ by $\hat{p}(x)$ defined as $\log \hat{p}(x) = \log q(x) - \int_0^1 r(x, s; \hat{\theta}_r) ds$, where $\log q(x)$ is given by the change-of-variable formula using the pre-trained model $F$ and the analytic expression of $\mathcal{N}(0, I_d)$. As a by-product, since our Q-flow net provides an invertible mapping $T_0^1$, we can use it to obtain an improved generative model on top of $F$. Specifically, the improved distribution $\tilde{Q} := (F \circ T_1^0)_\# \mathcal{N}(0, I_d)$, that is, we first use Q-flow to transport $\mathcal{N}(0, I_d)$ and then apply $F$. The performance of the improved generative model can be measured using the "bits per dimension" (BPD) metric, which is a widely used metric in evaluating the performance of generative models (Theis et al., 2015; Papamakarios et al., 2017). In our setting, the BPD can also be used to compare the performance of the DRE.

The results show that Ours reaches the improved performance in Table 1 against baselines: it consistently reaches smaller BPD than the baseline methods across all choices of $Q$. Meanwhile, we also note computational benefits in training: on one A100 GPU, Ours took approximately 8 hours to converge while DRE-$\infty$ took approximately 33 hours. In addition, we show trajectory of improved samples from $Q$ to $\tilde{Q}$ for RQ-NSF using the trained Q-flow in Figure 4a. Figure A.2 in the appendix shoes additional improved digits for all three specifications of $Q$.

## 4.5 Image-to-image translation

We use Q-flow to learn the continuous-time OT between distributions of RGB images of handbag (Zhu et al., 2016) and shoes (Yu & Grauman, 2014), which we denote as $P$ and $Q$ respectively.

Table 1: DRE performance on the energy-based modeling task for MNIST, reported in BPD and lower is better. Results for DRE-$\infty$ are from (Choi et al., 2022) and results for 1 ratio and TRE are from (Rhodes et al., 2020).

| Choice of $Q$ | RQ-NSF | | | | Copula | | | | Gaussian | | | |
|---|---|---|---|---|---|---|---|---|---|---|---|---|
| Method | Ours | DRE-$\infty$ | TRE | 1 ratio | Ours | DRE-$\infty$ | TRE | 1 ratio | Ours | DRE-$\infty$ | TRE | 1 ratio |
| BPD ($\downarrow$) | **1.05** | 1.09 | 1.09 | 1.09 | **1.14** | 1.21 | 1.24 | 1.33 | **1.31** | 1.33 | 1.39 | 1.96 |

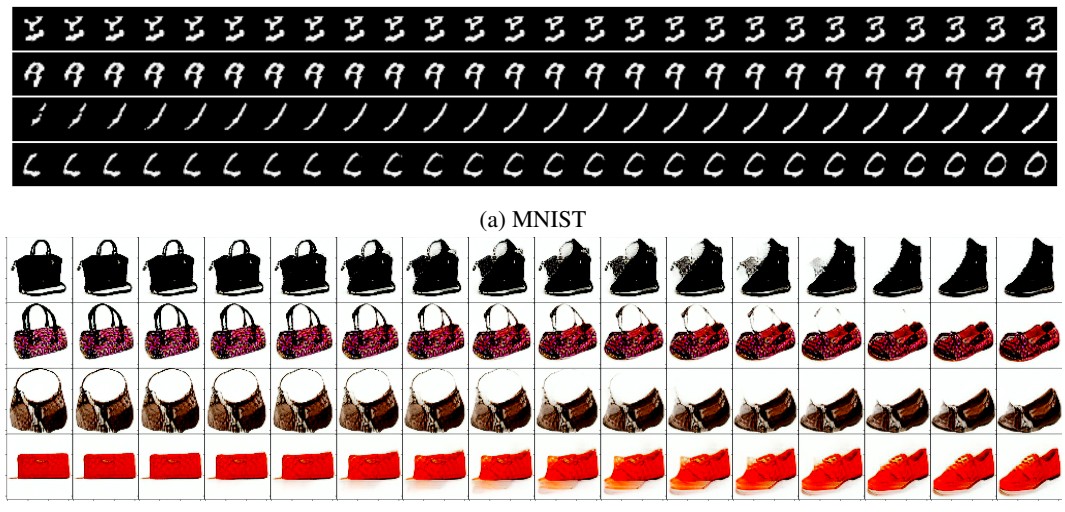

(a) MNIST

(b) Handbag → shoes

Figure 4: The trajectory of samples (in rows) from intermediate distributions of the Q-flow, as it pushes forward the base distribution (leftmost column) to the target distribution (rightmost column). Figure (a) shows the improvement of generated digits using the Q-flow. Figure (b) shows the image-to-image translation from handbag to shoes.

We follow the setup in (Korotin et al., 2023), where the goal of the image-to-image translation task is to conditionally generate shoe images by mapping test images of handbag through our trained Q-flow model. We train Q-flow in the latent space of a pre-trained variational auto-encoder (VAE) on $P$ and $Q$. Additional details are in Appendix B.4.

Figure 4b visualizes continuous trajectories from handbags to shoes generated by the Q-flow model. We find that Q-flow can capture the style and color nuances of corresponding handbags in the generated shoes as the flow model continuously transforms handbag images. Figure A.3 in the appendix shows additional generated shoe images from handbags. Quantitatively, we reach a Frechet Inception Distance ((Heusel et al., 2017), FID) of 15.95 between generated and true images of shoes. The FID remains competitive against FIDs from previous baselines, which range from 22.42 by DiscoGAN (Kim et al., 2017) to 13.77 by NeuralOT (Korotin et al., 2023). Meanwhile, since our Q-flow model learns a *continuous* transport map from source to target domains, it directly provides the gradual interpolation between the source and target samples along the dynamic OT trajectory as depicted in Figure 4b.

## 5 DISCUSSION

In this work, we develop Q-flow neural-ODE model that smoothly and invertibly transports between a pair of arbitrary distributions $P$ and $Q$. The flow network is trained to find the dynamic optimal transport between the two distributions and is learned from finite samples from both distributions. The proposed flow model shows strong empirical performance on simulated and real data for the tasks of density ratio estimation and image-to-image translation.

For future directions, first, the algorithm of training the Q-flow net can be further enhanced. Because the computational complexity scales with the number of time steps along the trajectory, more advanced time discretization schemes, like adaptive time grids, can further improve the computational efficiency which would be important for high dimensional problems. Second, there are many theoretical open questions, e.g., the theoretical guarantee of learning the OT trajectory, which goes beyond the scope of the current work. For the empirical results, extending to a broader class of applications and more real datasets will be useful.

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
