# OpenReview forum: "Computing high-dimensional optimal transport by flow neural networks"
_ICLR.cc/2024/Conference — Submitted to ICLR 2024_

### Official Review · Reviewer_qvyZ · 2023-10-24

**Soundness:** 4 excellent
**Presentation:** 4 excellent
**Contribution:** 2 fair
**Rating:** 5
**Confidence:** 3

**Summary:**

This paper proposes to learn the dynamic Optimal Transport trajectory between two distributions only known through samples. Authors propose to learn the velocity field by minimizing the dynamical OT problem where the marginals are enforced by minimizing the KL divergence. One of the main contribution of the paper is to propose a new density ratio estimation technique based on a logistic classification network. Then, the method is applied on several tasks, ranging from finding the trajectory between toy data, estimating the Mutual Information, Energy-based modeling and Image-to-Image translation.

**Strengths:**

Overall, the paper is well written and proposes a method to approximate the dynamic OT with Continuous Normalizing Flows. One of the main contribution is the new density ratio estimator which is shown to perform well compared to some baselines. The method is also demonstrated to work on several applications.

- A new Density Ratio Estimator used in order to approximate the dynamic OT, which is to the best of my knowledge original.
- Use a symmetric loss to better train the velocity field
- Different strategies to initialize the flow are discussed
- Several applications demonstrating the superiority of the method compared to others DRE estimators. Notably the experiments are mostly in high dimension.

**Weaknesses:**

- The comparisons seem to be made only with the same method using other density ratio estimation techniques. Other works which could be compared with could be e.g. [1] which propose a flow matching technique which can link arbitrary distributions.
- The Figures are not all of good quality. Notably, Figure 2 and 4 are a bit too small and we cannot really distinguish the results of Figure 2,b.

[1] Tong, Alexander, Nikolay Malkin, Guillaume Huguet, Yanlei Zhang, Jarrid Rector-Brooks, Kilian Fatras, Guy Wolf, and Yoshua Bengio. "Conditional flow matching: Simulation-free dynamic optimal transport." arXiv preprint arXiv:2302.00482 (2023).

**Questions:**

I think that some related works are not cited. For instance, [1] parameterize Normalizing Flows (NFs) with Monge maps, [2] train NFs using the JKO scheme and the dynamic formulation of OT and [3] improves the OT cost of Normalizing Flows. Also, [4] proposes a way to find a Normalizing Flow between two arbitrary distributions.


I found some other works which use Density Ratio Estimators based on Bregman divergences, e.g. [5, 6], and I am wondering whether these methods are competitive or not with the technique used in this paper.

Typos:
- Above equation (5): "The inner-loop training of $r_1$ is by"


[1] Huang, Chin-Wei, Ricky TQ Chen, Christos Tsirigotis, and Aaron Courville. "Convex potential flows: Universal probability distributions with optimal transport and convex optimization." arXiv preprint arXiv:2012.05942 (2020).

[2] Vidal, Alexander, Samy Wu Fung, Luis Tenorio, Stanley Osher, and Levon Nurbekyan. "Taming hyperparameter tuning in continuous normalizing flows using the JKO scheme." Scientific Reports 13, no. 1 (2023): 4501.

[3] Morel, Guillaume, Lucas Drumetz, Simon Benaïchouche, Nicolas Courty, and François Rousseau. "Turning Normalizing Flows into Monge Maps with Geodesic Gaussian Preserving Flows." arXiv preprint arXiv:2209.10873 (2022).

[4] Panda, Nishant, Natalie Klein, Dominic Yang, Patrick Gasda, and Diane Oyen. "Semi-supervised Learning of Pushforwards For Domain Translation & Adaptation." arXiv preprint arXiv:2304.08673 (2023).

[5] Feng, Xingdong, Yuan Gao, Jian Huang, Yuling Jiao, and Xu Liu. "Relative entropy gradient sampler for unnormalized distributions." arXiv preprint arXiv:2110.02787 (2021).

[6] Heng, Alvin, Abdul Fatir Ansari, and Harold Soh. "Generative Modeling with Flow-Guided Density Ratio Learning." arXiv preprint arXiv:2303.03714 (2023).

---

> ### Author Response · Authors · 2023-11-22
> **Response to Reviewer qvyZ**
>
> 1. **(Presentation) The Figures are not all of good quality. Notably, Figure 2 and 4 are a bit too small and we cannot really distinguish the results of Figure 2,b.**
>
> Thank you. We will re-make the figures in future revisions.
>
> 2. **(Citation) I think that some related works are not cited. For instance, [1] parameterize Normalizing Flows (NFs) with Monge maps, [2] train NFs using the JKO scheme and the dynamic formulation of OT and [3] improves the OT cost of Normalizing Flows. Also, [4] proposes a way to find a Normalizing Flow between two arbitrary distributions.**
>
>     [1] Huang, Chin-Wei, Ricky TQ Chen, Christos Tsirigotis, and Aaron Courville. "Convex potential flows: Universal probability distributions with optimal transport and convex optimization." arXiv preprint arXiv:2012.05942 (2020).
>
>     [2] Vidal, Alexander, Samy Wu Fung, Luis Tenorio, Stanley Osher, and Levon Nurbekyan. "Taming hyperparameter tuning in continuous normalizing flows using the JKO scheme." Scientific Reports 13, no. 1 (2023): 4501.
>
>     [3] Morel, Guillaume, Lucas Drumetz, Simon Benaïchouche, Nicolas Courty, and François Rousseau. "Turning Normalizing Flows into Monge Maps with Geodesic Gaussian Preserving Flows." arXiv preprint arXiv:2209.10873 (2022).
>
>     [4] Panda, Nishant, Natalie Klein, Dominic Yang, Patrick Gasda, and Diane Oyen. "Semi-supervised Learning of Pushforwards For Domain Translation & Adaptation." arXiv preprint arXiv:2304.08673 (2023).
>
> Thank you. We will add the citations in future revisions. Meanwhile, please see our response to the common questions above regarding the difference of our approach with theirs.
>
> 3. **Typos: Above equation (5): "The inner-loop training of r_1 is by”**
>
> Thank you. We will fix this in the revised draft in future revisions.

---

> > ### Comment · Reviewer_qvyZ · 2023-12-01
> >
> > I thank the authors for the response to the reviews. I would be encline to update my score if the revision includes comparisons with more baselines. For now, I will keep my score unchange.

---

### Official Review · Reviewer_3qrZ · 2023-10-27

**Soundness:** 2 fair
**Presentation:** 3 good
**Contribution:** 2 fair
**Rating:** 3
**Confidence:** 4

**Summary:**

In this paper, the authors utilize neural ODE to calculate optimal transport mapping in high-dimensional spaces. The proposed Q-flow model can learn a continuous invertible optimal transport. The Q-flow model is trained using a separate continuous-time neural network work classification loss along the time grid. Overall, this paper proposes a simple method to achieve learning optimal transport in high-dimensional space.

**Strengths:**

1. The proposed method is simple and effective
2. The writing is good and easy to follow

**Weaknesses:**

1. How the proposed two loss function satisfies the condition $\partial_t\rho+\nabla\cdot(\rho v)=0$ during training.
2. Does the proposed loss function affect the optimal transport between $P$ and $Q$? Maybe provide proof of achieving optimal transport via these two loss terms.
3. The author thinks bi-direction flow can achieve better numerical accuracy, but it seems there are no experiments to demonstrate this statement.
4. Any theoretical proof of bi-direction flow benefit will be better.
5. How does the KL loss impact the final training results (i.e., if the terminal condition is not considered, how does the final result become)?
6. No large-scale/high-resolution image generation experiments.
7. The authors are encouraged to compare their proposal with recent state-of-the-art diffusion based generation methods.

**Questions:**

Refer to the weakness section.

---

> ### Author Response · Authors · 2023-11-22
> **Response to Reviewer 3qrZ**
>
> 1. **(Continuity equation) How the proposed two loss function satisfies the condition $\partial_t \rho+\nabla \cdot (\rho v)=0$ during training.**
>
> The condition $\partial_t \rho+\nabla \cdot (\rho v)=0$ is the continuity equation which relates the density $\rho(x,t)$ to the velocity field $v(x,t)$. Specifically, if $x(t)\sim \rho(x,t)$ and $\partial_t x(t) = v(x,t)$, then $\rho(x,t)$ *will* satisfy the continuity equation. Therefore, we do not need to enforce this condition in practice; during training, we only need to focus on the boundary conditions (i.e., $\rho(x,0)=p(x), \rho(x,1)=q(x)$).
>
> 2. **(Loss to final result) How does the KL loss impact the final training results (i.e., if the terminal condition is not considered, how does the final result become)?**
>
> If the terminal condition is not considered, the learnt dynamic trajectory would not be between $P_0=P$ and $P_1=Q$, but between $P$ and $\hat{P}_1$, where $\hat{P}_1$ can differ from $Q$. In that case, this is different from our original goal of learning the dynamic OT between $P$ and $Q$ based on the BB-formula (see Eq (2)). In practice, as in the case of handbag -> shoes shown in Section 4.5, this would imply a poor match between the true shoes and generated ones, which are translated from handbags by the trained flow.
>
> 3. **(High-dim images) No large-scale/high-resolution image generation experiments.**
>
> Thank you. We will consider such examples in future revisions, besides the handbags -> shoes example in Section 4.5 on 64x64 images. We also want to emphasize that the main goal of the work is not to perform image generation, but to learn the dynamic OT leveraging the BB formula in Eq (2). Leveraging the learnt OT, we can thus effectively perform many other tasks such as DRE.
>
> 4. **(Against diffusion models) The authors are encouraged to compare their proposal with recent state-of-the-art diffusion based generation methods.**
>
> Thank you for the suggestion. We want to emphasize that our main goal is not to train a generative model, but to approximate the dynamic OT between arbitrary data distributions $P$ and $Q$. Doing so can be helpful in statistical inference tasks such as DRE. In comparison, diffusion models have reached SOTA performance on generative modeling (where $Q$ is typically the standard Gaussian), but these models are not learning the dynamic OT.

---

### Official Review · Reviewer_Q3SS · 2023-10-31

**Soundness:** 1 poor
**Presentation:** 3 good
**Contribution:** 1 poor
**Rating:** 3
**Confidence:** 4

**Summary:**

The authors propose a neural-ODE-based approach with min-max optimization for finding the solution of the dynamic optimal transport (OT) with the quadratic cost between two distributions with sample access. By using the flow-based training methodology, they perform optimization of the flow in both directions from the first distribution (initial) to the second (target) and vice versa. The method is applied to the density ratio estimation (DRE) problem on MNIST dataset, demonstrating improved results in comparisons with some baselines [6], [7]. The authors also apply their method to unpaired image-to-image translation on RGB data.

**Strengths:**

The proposed method outperforms methods [6] and [7] in the DRE problem on MNIST dataset.

**Weaknesses:**

- At the first glance, the authors position their main optimization objective as a minimization problem. However, having understood the paper better, one may realize that their objective actually constitutes the min-max optimization because it uses the variational (discriminator-based) estimation of KL divergence. It seems like the classification net can be viewed as a discriminator (in accordance with the formula (5) and Table 2 of the paper [9]). The trained flow-based network is used as a generator according to expressions (6) and (3). Unfortunately, the authors did not mention this important fact over the paper, which seems a little bit unfair with respect to the reader.
- The computation of integral of Neural-ODE along learned trajectories lies at the heart of computation inefficiency of the proposed algorithm in accordance with section 3.2. That is, the method is simulation-based.
- The authors demonstrate improved performance compared to [6],[7]  in the DRE problem in MNIST dataset. However, considering only the gray-scaled dataset MNIST, it is sufficiently difficult to argue that the proposed approach demonstrates significant enough improvement for this problem. So, I think it may be necessary to consider more high-dimensional and color datasets such as Celeba-64  and CIFAR-10 at least. Overall, it seems to me that the methodology for DRE which the authors use is not their method-specific. It seems like the classification network can be learned with any (e.g., trained with some other algorithm) generator. So it is not crystal clear what exactly the experiment on MNIST demonstrates.

Given the three prior weaknesses above, I wonder what are actual advantages which the current method provides compared to existing methods. For example, neural adversarial OT methods [10,11] are simulation-free. Existing flow-based methods [1],[2],[3],[4] are (usually) not simulation free but have simpler non-adversarial optimization. On top of each of these groups one seems to be able to learn DRE classifier networks. That is, it seems like the method proposed here combines disadvantages of two areas and overcomplicates the training process. So what is the reason to use this method in practice?

Also there are limited comparisons (both in terms of number of baselines and datasets) both with flow-based methods and adversarial OT methods. In particular, The authors of the article mention in related works 1.1 there are already many flow-based methods [1],[2],[3],[4]. Nonetheless, there are no comparisons with the aforementioned approaches in section 4.2 as well as 4.5. As for adversarial OT methods, there are only quick comparisons with [10] without any qualitatative analysis and only at 64x64 resolution

**Questions:**

- Why do you support training in both directions ? Which problems do we have while using the training of the flow-based network and classification net along only the forward trajectories ?

- The formula (7) seems to only be an upper-bound for the true Wasserstein-2 distance but not the exact distance, right?

- Since the proposed method is OT-solver with inserted flows, then it seems reasonable to test the approach on the benchmark [5] from the field.

**Papers:**

[1] - “Action matching: Learning Stochastic Dynamics from Samples”,  Neklyudov et al., 2022

[2] - “Building normalizing flows with stochastic interpolants”, Michael S. Albergo et al., 2023

[3] - “Flow matching for generative modeling”, Lipman et al., 2022

[4] - “Rectified flow: A marginal preserving approach to Optimal transport”, Liu Qiang, 2022

[5] - “Do neural optimal transport solvers work? A continuous Wasserstein -2 benchmark”, Korotin et al., 2021

[6] - “Telescoping Density-Ratio Estimation”, Rhodes et al., 2020

[7] - ”Density Ratio Estimation via Infinitesimal Classification”, Choi et al., 2021

[8] - “Density Ratio Estimation and Neyman Pearson Classification with Missing Data”, Givens et al., 2023

[9] - “f-GAN: Training Generative Neural Samplers using Variational Divergence Minimization”, Nowozin et al., 2016

[10] - ”Neural Optimal Transport”, Korotin et. al., 2022

[11] - “Neural Monge Map Estimation and Applications”, et. al. 2023

---

> ### Author Response · Authors · 2023-11-22
> **Response to Reviewer Q3SS**
>
> 1. **(Implicit variational learning) At the first glance, the authors position their main optimization objective as a minimization problem. However, having understood the paper better, one may realize that their objective actually constitutes the min-max optimization because it uses the variational (discriminator-based) estimation of KL divergence. It seems like the classification net can be viewed as a discriminator (in accordance with the formula (5) and Table 2 of the paper [9]). The trained flow-based network is used as a generator according to expressions (6) and (3). Unfortunately, the authors did not mention this important fact over the paper, which seems a little bit unfair with respect to the reader…[Meanwhile,] existing flow-based methods [1],[2],[3],[4]...have simpler non-adversarial optimization.**
>
>     [1] - “Action matching: Learning Stochastic Dynamics from Samples”, Neklyudov et al., 2022
>
>     [2] - “Building normalizing flows with stochastic interpolants”, Michael S. Albergo et al., 2023
>
>     [3] - “Flow matching for generative modeling”, Lipman et al., 2022
>
>     [4] - “Rectified flow: A marginal preserving approach to Optimal transport”, Liu Qiang, 2022
>
> Thank you for pointing this out. We will make this implicit variational learning clear in future revisions. We agree that compared to [1-4], we have to use a “discriminator” to enforce the boundary condition. However, there are essentially differences in terms of the objective and what we can achieve. First, [1-3] learn interpolations between $P$ and $Q$, but such interpolations are not necessarily the (dynamic) OT between $P$ and $Q$. In fact, such an interpolation can be used as the initialized flow for our OT refinement; we have considered using [2] as an initialization strategy in the handbag -> shoe example shown in Section 4.5. Regarding [4], the method can be viewed as an alternative direction descent of the BB formula (see Section 5.4 therein). However, the linear interpolation process in the alternative scheme “is *not* deterministic or ODE-inducible unless the fixed point is achieved”, making it essentially different from the BB approach that focuses on deterministic ODE solutions. In addition, there lacks empirical evaluation of the proposed rectified flow approach.
>
> 2. **(Simulation-based approach) The computation of integral of Neural-ODE along learned trajectories lies at the heart of computation inefficiency of the proposed algorithm in accordance with section 3.2. That is, the method is simulation-based**
>
> Thank you for pointing this out. We agree that the method is simulation-based. However, this is because we are directly trying to learn the dynamic OT based on the Benamou-Brenier formula (see Eq (2)). Prior works [Liu et al., 2022, Tong et al., 2023] have tried to learn the dynamic OT in a simulation-free manner, but as explained in the answers to the first common question, these approaches either fail to recover the exact deterministic ODE solution [Liu et al., 2022], or the method only works in theory with infinitely large batches [Tong et al., 2023].
>
> References:
>
> [Liu 2022] Rectified flow: A marginal preserving approach to Optimal transport]
>
> [Tong et al., 2023] Improving and generalizing flow-based generative models with minibatch optimal transport
>
> 3. **(W2 objective) The formula (7) seems to only be an upper-bound for the true Wasserstein-2 distance but not the exact distance, right?**
>
> Thank you for the question. As explained in the paper, “the population form of (7) in minimization can be interpreted as the discrete-time summed (square) Wasserstein-2 distance”, which is between densities $\rho(\cdot, t_{k-1})$ and $\rho(\cdot, t_k)$.

---

> > ### Comment · Reviewer_Q3SS · 2023-12-02
> > **Response to the answers**
> >
> > I thank the authors for providing some clarifications. It looks like there are a lot of additional discussions to include in the paper and various additional experiments to conduct. I think the paper is far from the publication condition and the authors themself seem to agree with this when write about "future revisions". I keep my score as is.

---

### Official Review · Reviewer_MAzo · 2023-11-05

**Soundness:** 3 good
**Presentation:** 3 good
**Contribution:** 3 good
**Rating:** 5
**Confidence:** 4

**Summary:**

The paper proposed an optimal transport flow method to transform images from one distribution to another distribution. The method trains a neural ODE mapping between two distributions. The loss function includes two parts: KL divergence and a Wasserstein-2 regularization, where the KL divergence relies on a pretrained classifier. The paper conducts experiments with toy data and also with real-world images to show that their method can generate high-quality flowed images.

**Strengths:**

1. The paper conducts several experiments showing that the flow can generate flow paths between two distributions.
2. The paper clearly describes the algorithm and the method. The writing is commendable.

**Weaknesses:**

1. In the training algorithm, there are two neural networks r0 and r1. That will add more complexity and difficulty in parameter tuning to the training scheme. It is a bit unclear on if one model is poorly trained, how would that affect the whole flow quality.

2. There are lot of metrics used in the experiment section: mutual information, FID, and BPD. If you can group them in one table or plot, it would be cleaner to compare the methods with all three metrics.

3. We recommend the authors cite the following two recent works on MMD and gradient flow:
Fan, J. and Alvarez-Melis, D., 2023. Generating synthetic datasets by interpolating along generalized geodesics. arXiv preprint arXiv:2306.06866.

Hua, X., Nguyen, T., Le, T., Blanchet, J. and Nguyen, V.A., 2023. Dynamic Flows on Curved Space Generated by Labeled Data. arXiv preprint arXiv:2302.00061.

**Questions:**

1. Have you done an ablation study on different loss functions or their weights?
2. In section 3.1, is it possible to use MMD in the loss instead of KL divergence?
3. Similar to question 2, is it possible to compute the KL divergence with the images themselves or embeddings of the images?

---

> ### Author Response · Authors · 2023-11-22
> **Response to Reviewer MAzo**
>
> 1. **(Presentation) There are a lot of metrics used in the experiment section: mutual information, FID, and BPD. If you can group them in one table or plot, it would be cleaner to compare the methods with all three metrics.**
>
> Thank you for the suggestion. We would do so in the future revisions, which will include additional experiments to verify the method.
>
> 2. **(Cite) We recommend the authors cite the following two recent works on MMD and gradient flow:**
> Fan, J. and Alvarez-Melis, D., 2023. Generating synthetic datasets by interpolating along generalized geodesics. arXiv preprint arXiv:2306.06866.
> Hua, X., Nguyen, T., Le, T., Blanchet, J. and Nguyen, V.A., 2023. Dynamic Flows on Curved Space Generated by Labeled Data. arXiv preprint arXiv:2302.00061.
>
> Thank you, we will cite these works in the future.
>
> 3. **(Different objective) In section 3.1, is it possible to use MMD in the loss instead of KL divergence?**
>
> Thank you for the question. We agree that the MMD objective can be an alternative relaxation of the terminal condition $\rho(\cdot, 1) =q$. We will explore the use of this objective in future works.
>
> 4. **(Ablation) Have you done an ablation study on different loss functions or their weights?**
>
> We performed an ablation study on the weight parameter $\gamma$ in Table A.2, where the DRE performance barely varies across $\gamma$. We will consider different loss functions (e.g., the MMD loss) in the future.
>
> 5. **(Additional evaluation) Similar to question 2, is it possible to compute the KL divergence with the images themselves or embeddings of the images?**
>
> Thank you for the question. We can estimate the KL divergence between generated images and true images using Eq (6). Specifically, assume we have trained a classification network $\hat r_1$ that is close to the optimal classifier $r_1^*$ between $P_1$ and $Q$. Then, the KL divergence between $P_1$ and $Q$ can be computed as $-\mathbb{E}_{x\sim P_1} \hat{r}_1(x)$, where the expectation is approximated by finite samples.

---

### Official Review · Reviewer_f3T8 · 2023-11-05

**Soundness:** 2 fair
**Presentation:** 2 fair
**Contribution:** 2 fair
**Rating:** 5
**Confidence:** 4

**Summary:**

The authors consider a problem of mapping one high-dimensional distribution to another using a flow in continuous time. To train the flow they proposed a loss function consisting of two terms. The first term is KL divergence between the given second distribution and the distribution, resulting from the flow, which takes the first distribution as a starting point. The second term can be interpreted as the discrete-time summed W2 distance. The authors used some existing approach to compute KL divergence and proposed an algorithm to estimate parameters of the flow. They demonstrated on a number of examples that the proposed approach provides good estimate of log density ration, and also can provide nicely-looking flows with good FID values in case of images.

**Strengths:**

- sufficiently clearly written paper
- natural idea of the algorithm
- detailed description of the algorithm and experimental study
- discussion of the features of the computational implementation of the algorithm
- interesting practical results

**Weaknesses:**

- the title of the paper is "Computing high-dimensional optimal transport by flow neural networks". However, a significant part of the paper is devoted to benchmarking of the capability of the algorithm to perform density ratio estimation. So it is not clear what is the main aim of the paper - to compute OT, or to estimate log density ratio

- If the main aim is DRE, then it is necessary to provide detailed comparison with other DRE methods, as there are many papers on this topic. E.g., what is the difference of the proposed approach with the approach https://openreview.net/forum?id=kOIaB1hzaLe

- it is not clear why the proposed algorithm estimates optimal transport

- experimental results to verify efficiency of computed W2 high-dimensional optimal transport are not enough to claim accuracy and efficiency of the proposed approach. E.g. the authors consider some image translation tasks, but FID score used to characterise accuracy does not guarantee that the computed W2 high-dimensional optimal transport map is accurate.

**Questions:**

- it is not clear how to tune a value of gamma in (3). Any recipes for automatic tuning?

- page 9: "Meanwhile, since our Q-flow model learns a continuous transport map from source to target domains, it directly provides the gradual interpolation between the source and target samples along the dynamic OT trajectory as depicted in Figure 4b."

Any comments on why the trajectory corresponds to OT trajectory? To construct a flow the authors optimise (3), which contains two terms, and it is not clear why such optimisation formulation guarantees any optimality or that the mapped distribution coincides with the second distribution q.

- since the authors claim they compute W2 optimal transport, it is important to benchmark their approach on problems with ground truth solutions. There exist such benchmark, see https://github.com/iamalexkorotin/Wasserstein2Benchmark (Do Neural Optimal Transport Solvers Work? A Continuous Wasserstein-2 Benchmark, NeurIPS 2021)

- page 18 (the second line after the displayed formula 15): Why Q = N(0,I_d) if P = N(0,Sigma)?

- page 19: it is not clear how gamma = 0.5 was selected. Why not 0.6?

---

> ### Author Response · Authors · 2023-11-22
> **Response to Reviewer f3T8**
>
> 1. **(contribution, OT or DRE) the title of the paper is "Computing high-dimensional optimal transport by flow neural networks". However, a significant part of the paper is devoted to benchmarking the capability of the algorithm to perform density ratio estimation. So it is not clear what is the main aim of the paper - to compute OT, or to estimate log density ratio**
>
> Thank you for the question. We intend to present methods that can be useful for both the OT problem and the DRE task. Specifically, we aim to approximate the dynamic OT solution in BB formula (see Eq. (2)) by a flow model, where the learnt continuous OT trajectory enables more accurate DRE. In future revision, we will make our contribution more clear with better experimental design.
>
> 2. **(Not showing OT through FID) experimental results to verify efficiency of computed W2 high-dimensional optimal transport are not enough to claim accuracy and efficiency of the proposed approach. E.g. The authors consider some image translation tasks, but the FID score used to characterize accuracy does not guarantee that the computed W2 high-dimensional optimal transport map is accurate.**
>
> Thank you for pointing this out. We agree that a low FID score does not indicate an accurate learnt OT trajectory; we considered this example mainly to illustrate the scalability of our approach to non-trivial high-dimensional image datasets. In the future, we will more quantitatively evaluate our learnt OT trajectory, such as through this benchmark [Korotin et al., 2021]
>
> Reference:
>
> [Korotin et al., 2021] Do Neural Optimal Transport Solvers Work? A Continuous Wasserstein-2 Benchmark
>
> 3. **($\gamma$ choice) it is not clear how to tune a value of gamma in (3). Any recipes for automatic tuning? On page 19: it is not clear how gamma = 0.5 was selected. Why not 0.6?**
>
> Thank you for the question. At present, we don’t have a recipe for automatic tuning, but find that using $\gamma=0.5$ is typically a good starting point. We will explore such recipes in the future. Nevertheless, In Table A.2, we conducted an ablation study on the MNIST example with respect to $\gamma$, where we verified that the performance is not sensitive to the choice of $\gamma$.
>
> 4. **(Clarification) page 18 (the second line after the displayed formula 15): Why Q = N(0,I_d) if P = N(0,Sigma)?**
>
> Thank you for the question. This example is taken from [Rhodes et al., 2020, Appendix E]. Specifically, we note that $\Sigma$ is a block-diagonal matrix with 2x2 blocks. As a result, for $X\sim N(0,\Sigma)$, $U=(x_1,x_3,\ldots,x_{d-1})$ and $V=(x_2,x_d,\ldots,x_d)$ are random vectors drawn from $N(0,I_{d/2})$. As a result, $p(U)p(V)=Q(x)$ for $Q=N(0,I_d)$.
>
> Reference:
>
> [Rhodes et al., 2020] Telescoping Density-Ratio Estimation

---

### Author Response · Authors · 2023-11-22
**Response to common questions**

Thank you to all reviewers for the valuable feedback and constructive comments. Common concerns raised include (1) the comparison of our proposed OT method with other methods, (2) clarifications regarding the method's design, and (3) the necessity for additional experiments and comparisons. The additional questions and comments of each reviewer are addressed in the specific responses below. R1 = Reviewer f3T8, R2 = Reviewer MAzo, R3 = Reviewer Q3SS, R4 = Reviewer 3qrZ, R5 = Reviewer qvyZ

1. **(R3, R5) Relation/advantage over other OT methods, some of which claim to learn dynamic OT as well [Tong et al., 2023].**

    Paper list:
- [Huang et al., 2021] Convex Potential Flows: Universal Probability Distributions with Optimal Transport and Convex Optimization.
- [Liu 2022] Rectified flow: A marginal preserving approach to Optimal transport.
- [Morel et al., 2023] Turning Normalizing Flows into Monge Maps with Geodesic Gaussian Preserving Flows.
- [Fan et al., 2023] Neural Monge Map estimation and its applications
- [Korotin et al., 2023] Neural Optimal Transport
- [Tong et al., 2023] Improving and generalizing flow-based generative models with minibatch optimal transport


We appreciate the suggested related works by the reviewers, which will be cited in our future revisions. These works fall into two categories: (1) addressing static OT for Monge map [Huang et al., 2021, Morel et al., 2023, Fan et al., 2023, Korotin et al., 2023] and (2) tackling dynamic OT for continuous-time velocity field [Liu 2022, Tong et al., 2023]. We will begin by summarizing the key ideas of these works and then discuss the connections and differences between our work and theirs.
* **Static OT**: The problem involves finding a one-step function $T$ (Monge map), such that $T_{\\#}P = Q$, upon minimizing $\mathbb{E}_{x\sim P} [c(x,T(x))]$ for a given cost function $c$. [Huang et al., 2021] utilizes the input convex neural network (ICNN) to learn the Monge map, avoiding the need to solve the original Monge problem. [Morel et al., 2023] suggests transforming trained normalizing flows for enhanced OT efficiency, based on Brenier's polar factorization theorem. [Fan et al., 2023] formulates the problem as a saddle point problem, accommodating general cost functions. [Korotin et al., 2023] applies the dual formulation to enable the training of a "weak (static) OT," resulting in an optimal map $T(x)=\pi(\cdot|x)$ that produces a conditional distribution given $x$.
* **Dynamic OT**: The problem aims to find the optimal velocity field $v(x(t),t)$ to convert $P$ into $Q$ (refer to Eq (2) in our paper). To achieve this, [Liu 2022] introduces rectified flow, which can be viewed as an alternate direction descent approach for this problem (see [ibid., Section 5.4]). In contrast, [Tong et al., 2023] approximate the dynamic OT solution by initially solving the static OT problem using mini-batch OT.

The equivalence of static and dynamic OT formulations with the Euclidean cost $c(x,y)=||x-y||$ was initially demonstrated in [Benamou & Brenier, 2000]. However, for tasks like DRE that demand a continuous trajectory from $P$ to $Q$, the dynamic formulation can offer a more direct and precise solution compared to the static approach. The static method necessitates additional interpolation between $x$ and $T(x)$ to construct the trajectory, which might not align with the dynamic OT solution in practice if $T$ is sub-optimally trained.

Our proposed method is closely related to the second line of work, which aims to address the dynamic OT problem (Eq (2)). We note certain limitations in [Liu 2022, Tong et al., 2023]. First, the alternating scheme in [Liu 2022, Section 5.4] may not recover the deterministic ODE solution for (2), as it alternates between a deterministic flow and a linear interpolation process which is *not* deterministic or ODE-inducible unless the fixed point is achieved. On the other hand, [Tong et al., 2023] first solves the static OT problem between mini-batches of samples and then learns the velocity field. In theory, the approach only works as the batch size approaches infinity, which is impractical in practice. In contrast, our approach involves discretizing (2) with a relaxed terminal condition to learn the velocity field driving the ODE, resulting in a simpler and more direct approach.

Reference:

[Benamou & Brenier, 2000] A computational fluid mechanics solution to the Monge-Kantorovich mass transfer problem

---

> ### Author Response · Authors · 2023-11-22
> **Response to common questions (cont.)**
>
> 2. **Method**
> - **(R2, R3, R4) Why do we have bi-directional design? Cannot one just train along one direction?**
> - **(R1, R4) Why does the design of loss objective learn the dynamic OT? Any guarantees?**
>
> **(Response to the first bullet)** In practice, introducing bidirectional training enhances numerical accuracy and stability. We acknowledge the added computational cost of bidirectional training compared to training in one direction. In the future, we'll explore methods to enhance training efficiency and potentially simplify the design.
>
> **(Response to the second bullet)** We based our loss objective on the Benamou-Brenier formula (see Eq (2)), which comprises the KL term in (6) and the transport cost term in (7). It's worth noting that (6) relaxes the terminal condition by employing a classifier, while (7) can be seen as the discrete-time sum of the (square) Wasserstein-2 distance.
>
> We intuitively explain why minimizing (3) leads to learning dynamic OT. Firstly, when we train the classifier $r_1$ near optimality, minimizing the expected value of (6) enforces the terminal condition. Secondly, with a sufficiently fine time grid $0=t_0<t_1 < \ldots < t_K=1$, the expected value of (7) approximates the continuous-time transport cost well. We plan to formalize this intuition in future revisions.
>
> 3. **Experiment**
> - **(R1, R3, R5) More comparison with DRE baselines and/or examples. MNIST is not convincing (R3), and wants larger datasets (if DRE on images).**
>
>     Paper list:
>
>     [Feng et al., 2021] Relative entropy gradient sampler for unnormalized distributions
>
>     [Miller et al., 2022] Contrastive Neural Ratio Estimation
>
>     [Heng et al., 2023] Generative Modeling with Flow-Guided Density Ratio Learning
>
>     [Givens et al., 2023] Density Ratio Estimation and Neyman Pearson Classification with Missing Data
>
> We thank the reviewers for the suggestion. We will compare our proposed OT+DRE method against these baselines on larger datasets in future revisions. Below, we summarize the key ideas of these methods and highlight the difference between what we proposed and theirs.
>
> Some of these DRE baselines are based on Bregman divergence [Feng et al., 2021, Heng et al., 2023] while others are based on cross entropy loss [Miller et al., 2022]. The work by [Givens et al., 2023] further extends DRE under the Kullback-Leibler importance estimation procedure when missing data are present, but the DRE estimator is not based on neural networks.
>
> We discovered that existing baselines only focus on learning a *single-step* estimator that optimally recovers $\log(p/q)$. However, previous studies [Rhode et al. 2020, Choi et al. 2022] have shown that DRE's empirical performance can significantly degrade when the KL-divergence between $p$ and $q$ exceeds tens of nats. This has prompted us to adopt a different approach: constructing an OT trajectory between $p$ and $q$ before applying DRE, which is continuous in time (see Appendix A). Empirically, this approach has resulted in improved performance compared to both the single estimator and telescopic estimators using non-OT trajectories.
>
> References:
>
> [Rhode et al., 2020] Telescoping Density-Ratio Estimation
>
> [Choi et al., 2022] Density Ratio Estimation via Infinitesimal Classification
>
>
> - **(R1, R3, R5) More comparison with OT baselines. For example, this benchmark https://github.com/iamalexkorotin/Wasserstein2Benchmark.**
>
> We thank the reviewers for the suggestion. We will compare our proposed method against these OT baselines in future revisions, especially on the high-dimensional examples of Celeb-A dataset.

---

### Meta-Review · Area_Chair_LvRH · 2023-12-08

**Metareview:**

The authors propose a flow-based model for dynamic optimal transport (OT) between two continuous distributions. The authors show advantages of the proposed method on several tasks including mutual information estimation, generative models, image-to-image translation. The Reviewers appreciate the novelty of the proposed method, but also raise concerns about related work, quality of the learned OT trajectories, as well as related baselines. In brief, I think the submission is not ready for publication yet. The authors can follow the Reviewers' comments and suggestions to improve the work.

**Justification For Why Not Higher Score:**

+ The Reviewers raised several issues, e.g., related work, quality of the learned OT trajectories, as well as related baselines. In the rebuttal, the authors also agree to take into account these raised concerns to improve the work.

+ The proposed idea is interesting, there are many rooms to improve it more. It is better to improve the work more from those raised concerns.

**Justification For Why Not Lower Score:**

N/A

---

### Decision · Program_Chairs · 2024-01-16

Reject